# Examining the Effects of the Pandemic on Entrepreneurial Activities among Urban Single Mothers: An Exploratory Study

**Abdullah Sallehhuddin Abdullah Salim [1], Norzarina Md Yatim [2,*] and Salmi Md Zahid [2]**

1   Faculty of Accountancy and Management, Universiti Tunku Abdul Rahman, Kajang 43000, Malaysia; sallehhuddin@utar.edu.my
2   Faculty of Management, Multimedia University, Cyberjaya 63100, Malaysia; salmi.md.zahid@mmu.edu.my
*   Correspondence: norzarina.yatim@mmu.edu.my

**Abstract:** This study was conducted in Malaysia to examine the effectiveness of the microfinance programme for urban single mother entrepreneurs in MSMEs during the COVID-19 pandemic and Movement Control Order (MCO). Implemented as a response to the pandemic, the MCO significantly disrupted businesses, particularly MSMEs. The study aimed to investigate the relationship between empowerment factors (economic, social, digital, and psychological) and governance aspects concerning the effectiveness of microfinance programmes. Using a positivist paradigm and employing quantitative methods through online questionnaire distribution, this research established a framework based on empowerment theory. The findings underscore the importance of economic empowerment, digital empowerment, and governance aspects for microfinance programme success, and provide empirical backing for suitable mitigation strategies for MSME entrepreneurs. The study emphasises the importance of supporting single mother entrepreneurs through various developmental activities, technical and vocational training, and comprehensive financial and non-financial aid initiatives. It stresses the critical role of women, particularly single mothers, in propelling societal and economic advancement, and advocating for their empowerment through targeted interventions. Overall, the findings enhance understanding of the challenges MSMEs face during crises, and offer insights for policymakers and microfinance agencies to strengthen support for single mother entrepreneurs in navigating future challenges and fostering economic resilience and development.

**Keywords:** microfinance; single mother entrepreneurs; empowerment; urban; pandemic COVID-19

## 1. Introduction

The Malaysian government was one of the first countries in Asia to announce the Movement Control Order (MCO), which began on 18 March 2020 as a precautionary measure and a response approach to curb the spread of the COVID-19 pandemic. Among the restrictions implemented was the closure of government offices and private premises, except for those related to essential services such as health and safety, telecommunications, retail, finance, and transportation. Although the MCO was able to curb the spread of the virus, its enforcement left a negative impact on the global economy, industry, companies, and micro, small and medium enterprises (MSMEs).

For MSMEs, the pandemic had a significant impact on business activities. Among the obstacles MSME entrepreneurs faced included problems in cash flow due to loss of daily income, operational disruptions, layoffs, and supply chain disruptions (Che Omar et al. 2020; Fabeil et al. 2020). In addition, the main obstacle was the disruption of business operations and some enterprise sectors, which ceased permanently due to financial problems from the effects of temporary closures during the restrictions (Bartik et al. 2020). In addition, Fabeil et al. (2020) also found that small businesses in rural areas experienced more significant challenges than enterprises in urban areas, and rapidly deteriorated due to their remoteness, especially in terms of infrastructure constraints, labour availability, and

limited financial reserves. Although governmental and non-governmental organisations provide various efforts and assistance, MSMEs have different experiences in facing the COVID-19 pandemic, and an uneven ability to reduce the impacts of the MCO on their business activities. Furthermore, the findings of Cook (2015) show that 75 per cent of businesses without a continuity plan will collapse within three years post-crisis, and Bartz and Winkler (2016) emphasise that MSMEs require a longer regrowth period following a crisis than large companies that grow faster and are more flexible.

However, most available studies of MSMEs focused on the impact of the COVID-19 pandemic alone; hence, the mitigation approach to reduce the effect has yet to be fully explored, including the factors that may affect the effectiveness of the mitigation strategy. Therefore, this study aimed to ascertain the relationship between empowerment factors and governance aspects on the effectiveness of microfinance programmes in helping single mother MSME entrepreneurs deal with the pandemic crisis in emerging economies like Malaysia. Specifically, this study sought answers to the main research question—*What is the relationship between the four indicators of empowerment of single mother MSME entrepreneurs—economic, social, digital, and psychological—as well as the governance aspects and the effectiveness of microfinance programmes when facing the COVID-19 pandemic and MCO?*

## 2. Literature Review

### 2.1. Theoretical Foundation of the Study

This study uses the empowerment theory to explain the relationship between empowering women MSME entrepreneurs and achieving effective microfinance programmes in addressing the challenges of the COVID-19 pandemic. According to Swift and Levin (1987), empowerment theory encompasses the link between processes and outcomes. It explains that structures, actions, resources, activities, or programmes may be empowering, and that the outcomes of such processes result in a level of being empowered. Empowering processes are one's efforts or attempts to gain control, obtain required resources, and apply the understanding of the external environment in becoming a problem solver or decision maker; then, the empowered outcomes refer to the consequences of one's efforts or attempts to gain greater control, or the effects of intervention-designed programmes (Zimmerman 1995). Furthermore, Zimmerman (1995) believes one can take the empowerment process at individual, organisational, and community levels.

The theory of empowerment includes various dimensions, forms, contexts, and units of analysis, and one of them is the process of women's empowerment (Rahman et al. 2018). Women empowerment enables women to become self-sufficient and independent, have a positive self-image, and control or obtain resources; hence, the empowerment process results in a life of well-being (Hashim et al. 2023). A study by Addai (2017) explains that women's empowerment in multiple dimensions, such as economic, social, legal, and psychological, has positively impacted micro-financing programmes. Similarly, Ahmad et al. (2019) found support for the relationship of economic, social, and psychological empowerment and management aspects towards the effectiveness of microfinance programmes. Furthermore, in response to COVID-19 challenges, studies like Sri Rahayu et al. (2023), Mohammad et al. (2022), Hamdan et al. (2021), and Che Omar et al. (2020) have urged the mastery of digital competencies to survive the post-pandemic era.

### 2.2. COVID-19 Pandemic Impact on MSMEs

A crisis can be defined as a situation faced by an individual, group or organisation that results in disruption to management using standard routine procedures (Anthony et al. 2019; Lai 2020; Ran et al. 2020), and is categorised into three types, which are gradual threats, periodic threats, and sudden threats (Booth 1993). In the context of this study, the COVID-19 pandemic crisis is categorised as a 'sudden threat' that occurs unexpectedly, which not only affects health conditions, but also causes global economic shock. A survey by Bartik et al. (2020) found that 5800 small businesses in America are facing a fragile financial situation due to the COVID-19 pandemic. The findings of the study found that the median of firms

with monthly expenses below $10,000 had cash in hand that lasted for one month, whereas in pre-pandemic or crisis times, the median for similar firms had cash on hand for less than 15 cash days given the higher level of expenses. This situation reflects that MSMEs are spending less during the crisis, which is unsuitable for the firm and the economy in the long term. In addition, the outbreak also caused adverse effects on layoffs and supply chain disruptions. For example, a study by Che Omar et al. (2020) found that MSMEs needed help in obtaining raw materials, which, on average, were mostly imported from China, and the number of available suppliers was becoming smaller or limited. Meanwhile, Fairlie (2020) analysed the pandemic's impact on active small businesses in the United States using national data from April 2020. The findings revealed that the African-American business community experienced a 41 per cent drop, while Latino business owners declined by 32 per cent, and Asian business owners decreased by 26 per cent.

*2.3. Microfinance Programme for Single Mother MSME Entrepreneurs in Malaysia during the COVID-19 Pandemic*

For nearly four decades, microfinance has become a key element in economic development strategies worldwide, especially in emerging and developing countries like Malaysia. The Amanah Ikhtiar Malaysia (A.I.M.) was launched in 1987, and is a federal government agency that manages microfinance programmes to improve vulnerable groups' well-being. Microfinance facility refers to the provision of loans and credit facilities where the amount offered is relatively small, and these facilities are developed to meet the capital or cash needs of poor and needy communities, including MSMEs. Microfinance is also a financial aid to small traders to eradicate poverty and change the standard of family life (Rahman et al. 2018). The theory of empowerment is an idea that is growing now in the context of advancing the socioeconomics of microfinance programme participants (Nazier and Ramadan 2018).

In dealing with the COVID-19 pandemic, through a circular with the reference number AIM/JPU/300-02/01(01) dated 24 March 2020, A.I.M. launched an economic stimulus package amounting to MYR 682,357,904.00 to help A.I.M. participants, also known as Sahabat A.I.M., to overcome the challenges of the pandemic and MCO. The economic stimulus package involves (1) the postponement of the collection of weekly repayments or a moratorium involving 373,815 Sahabat A.I.M., which involves a financial implication of MYR 555 million; (2) authorization to withdraw compulsory savings (S.W.) on a one-off or one-off basis of a maximum of MYR 300 to 373,815 Sahabat A.I.M. involving a financial implication of MYR 112.5 million; (3) deferment of I-Lestari and I-Usahawan Koop Sahabat installments involving 400 Sahabat A.I.M. involving a sum of MYR 300,000; (4) postponement of the Sahabat Ar-Rahnu auction or Islamic mortgage until the end of the MCO involving 1500 Sahabat A.I.M. with a financial implication of MYR 120,000; (5) a group fund loan (P.T.K.) offering involving 10,000 Sahabat A.I.M. with a maximum limit of MYR 1000 per borrower, with a financial implication of MYR 10 million; (6) distribution of cash donations amounting to MYR 250 to 17,444 Sahabat A.I.M. involving financial implications of MYR 4,361,104 channeled through Business Zakat Sahabat Koop MYR 621,104, A.I.M. Welfare Assistance Fund MYR 2.6 million, and A.I.M. Corporate Benefit Fund MYR 1.14 million; and (7) the organisation of Sahabat A.I.M. Entrepreneurship Workshop 2020, which emphasises aspects of online business or the digital economy involving 200 Sahabat A.I.M. (A.I.M. 2020a, 2020b).

Meanwhile, Hamdan et al. (2021) explored the impact of the COVID-19 pandemic on MSME entrepreneurs under the A.I.M. microfinance programme and the approaches used to mitigate the effects of COVID-19 on the participants' businesses. This study used a qualitative approach, in-depth interviews with six female micro-entrepreneurs in a semi-structured interview format, selecting respondents through a purposive sampling technique utilising the A.I.M. e-commerce platform known as Bazar Sahabat. The study provides two main themes for impact on micro-entrepreneurs: financial issues and operational disruptions. As for the mitigation approach, the consensus pattern among respondents

is related to adopting new norms, including changing the location of business operations from physical stores to home-based and online platforms.

### 2.4. Effectiveness of Microfinance Programme

Ahmad et al. (2019) examined the effectiveness of the microfinance programme managed by Hijrah—a Selangor state government agency. The study found that the programme's level of effectiveness is high. To measure the level of effectiveness, the researchers used five indicators consisting of the following: (1) respondents felt that the Hijrah Selangor microfinance programme should be participated by all single mothers; (2) the Hijrah Selangor microfinance programme helps individuals in eradicating poverty; (3) this programme is effective in increasing income; (4) this programme can change lives; and (5) this programme is successful in addressing the issue of poverty among single mothers. In addition, Koh et al. (2021) studied the factors that influence the effectiveness of the A.I.M. microfinance programme in helping to address the issue of urban poverty. This study measures effectiveness through two indicators—participant income and participant welfare.

### 2.5. Economic Empowerment

The availability of microfinance facilities is critical to overcoming the financial constraints identified as one of the leading causes preventing the growth and sustainability of MSMEs in developing countries (Wellalage and Locke 2017). Therefore, credit facilities play an essential role in facilitating the flow of capital, which can increase an individual's economic growth (Mariyono 2019). Besides, microfinance can also help to increase business in the real sector and stimulate economic growth, lower the unemployment rate through increased labour demand, increase income, and lower the poverty rate (Sipahutar et al. 2016). In addition, microfinance empowers female borrowers to make household decisions related to mobility, daily expenses, children's schooling, and health expenses (Al-shami et al. 2017a). Next, Al-shami et al. (2017b) studied the effect of productive loans provided by A.I.M. on the welfare and empowerment of women's households. It was found that microfinance facilities significantly and positively affect the borrower's household income and personal asset ownership. Furthermore, to overcome COVID-19 pandemic challenges, various assistance programmes were formulated by the Malaysian government, like the PRIHATIN package, to empower MSME entrepreneurs economically and financially (Mohammad et al. 2022; Hamdan et al. 2021; Che Omar et al. 2020). Therefore, the first hypothesis statement of the study is the following:

**H₁:** *There is a positive relationship between economic empowerment and the effectiveness of microfinance programmes.*

### 2.6. Social Empowerment

According to Walid and Peng (2022), entrepreneurial sustainability depends on social relationships; using social relationship failure as a dummy, the study found a significant relationship between both variables. In addition, studies by Zaleskiewicz et al. (2020) and Yukongdi and Lopa (2017) found that the relationship of one to family members, communities, societies, and workplace colleagues can determine the ups and downs of entrepreneurial efforts. Furthermore, a study by Ilieva-Koleva and Dobreva (2021) highlights the contribution of social empowerment, which is people-centred, e.g., customers, employees, communities, and other relevant stakeholders towards entrepreneurship activities in the form of corporate social responsibilities, social enterprises, non-profit organisations, and cooperatives. In addition, a study by Ahmad et al. (2019) found evidence of the relationship between social empowerment and the effectiveness of microfinance programmes by measuring one's ability to manage relationships with others. Hence, the second hypothesis statement of the study is as follows:

**H₂:** *There is a positive relationship between social empowerment and the effectiveness of microfinance programmes.*

## 2.7. Psychological Empowerment

Meng and Sun (2019) define psychological empowerment as "an intrinsic task motivation reflecting a sense of self-control concerning one's tasks, efforts, assignments, jobs, and works". In addition, Guberina et al. (2023) relate psychological concepts as "appraisal of one's attributes to their personal, social, and workplace status or self-image". In addition, Ab-Rahim et al. (2018) express psychological empowerment in three forms of power, namely (1) social power—awareness, knowledge, and ability to participate in society and access to resources without any form of discrimination; (2) political power—freedom and involvement in the decision-making process; and (3) psychological power—the potential of individuals and their judgments that influence social and political power. In their study, Ahmad et al. (2019) found evidence of the relationship between psychological empowerment and the effectiveness of microfinance programmes by measuring individual thinking and behavioural patterns on self-confidence, self-skills, self-esteem, and self-control. Based on the preceding arguments, the third hypothesis statement of the study is the following:

**H₃:** *There is a positive relationship between social empowerment and the effectiveness of microfinance programmes*

## 2.8. Digital Empowerment

In addition to educational issues, Hamdan et al. (2021) also dove into the digital challenges of the COVID-19 pandemic and the MCO against MSME entrepreneurs. Specifically, for the operational aspect, the main challenge faced by respondents is using digital platforms to continue business activities such as promotion and sales. The study also explains the need for regulators and microfinance scheme operators to conduct more training to build the capacity and skills of respondents to master the digital platform and prepare for future shocks. Furthermore, Che Omar et al. (2020) explored the survival strategies of MSME entrepreneurs during the COVID-19 pandemic, and found support for using digital technology to spearhead marketing and publicity efforts. Sri Rahayu et al. (2023) supported a similar point of view in the context of MSME entrepreneurs in Indonesia optimising digital platforms like social media tools to reach potential and existing customers. In addition, Mohammad et al. (2022) underscore the importance of digital technology adaptation among MSME entrepreneurs in preserving their businesses from the impacts of the COVID-19 pandemic. Meanwhile, Lythreatis et al. (2021) systematically reviewed and synthesised 50 articles from 24 countries about the digital divide. The study consolidates three levels of the digital divide covering Level 1: access to devices and an internet connection; Level 2: digital competence, skills, and literacy; and Level 3: the ability to use digital competence for life mobilisation. The study also classified the factors affecting the digital divide into nine themes: sociodemographic, socioeconomic, personal elements, social support, types of technology, digital skills, rights, infrastructure, and large-scale phases of life. Therefore, the fourth hypothesis statement of the study is the following:

**H₄:** *There is a positive relationship between digital empowerment and the effectiveness of microfinance programmes.*

## 2.9. Management and Governance of Microfinance Programmes

Ahmad et al. (2019) assessed the impact of management and monitoring on the effectiveness of the Hijrah microfinance programme offered by the Selangor state government. The study used nine indicators to measure management and monitor variables. The relevant indicators are the following: (1) Hijrah officers provided exemplary service without discrimination; (2) the effectiveness of the training provided by Hijrah Selangor; (3) satisfaction with the service provided; (4) good feedback from Hijrah officers; (5) confidence in the experience possessed by Hijrah officers; (6) confidence that Hijrah officers are given sufficient training in understanding the needs of aid recipients; (7) accessible complaint channels; (8) swift action when needed; and (9) frequency of monitoring by Hijrah officials. The analysis of the study found that the indicator with the highest mean

score was the excellent treatment by the officers without discrimination. Overall, the study found that the management level of Hijrah programme monitoring in Selangor is moderate. In addition, Koh et al. (2021) also examined aspects of training conducted by A.I.M. for microfinance participants in the Federal Territories of Kuala Lumpur, Penang, and Johor. The study found that the training managed by A.I.M. significantly improved the participants' socioeconomic welfare. The analysis also found that managed training indirectly enhances the socioeconomics of microfinance participants, with increased income. The management aspect of microfinance participant training and its importance to the effectiveness of the microfinance programme was also confirmed by Al Mamun et al. (2018). Hence, the fifth hypothesis statement of the study is as follows:

**H₅:** *There is a positive relationship between management and governance and the effectiveness of microfinance programmes.*

### 3. Research Methods

#### 3.1. Research Design and Research Framework

This study adopted a positivist paradigm. It deployed a quantitative method through questionnaire distribution, which is consistent with the methodology of previous studies of a similar nature. A research framework was formed based on the results of previous studies related to factors that contribute to the effectiveness of microfinance programmes (Al-shami et al. 2014), as well as considering the background of crises such as epidemics and movement restriction orders (Mohammad et al. 2022; Hamdan et al. 2021). The focus of this study is to examine the relationship between indicators of empowerment of single mother entrepreneurs participating in microfinance and the effectiveness of A.I.M.'s microfinance programme when dealing with the COVID-19 pandemic and the MCO. Figure 1 illustrates the antecedents influencing the effectiveness of microfinance programs amidst the COVID-19 pandemic. The framework of the study is as follows:

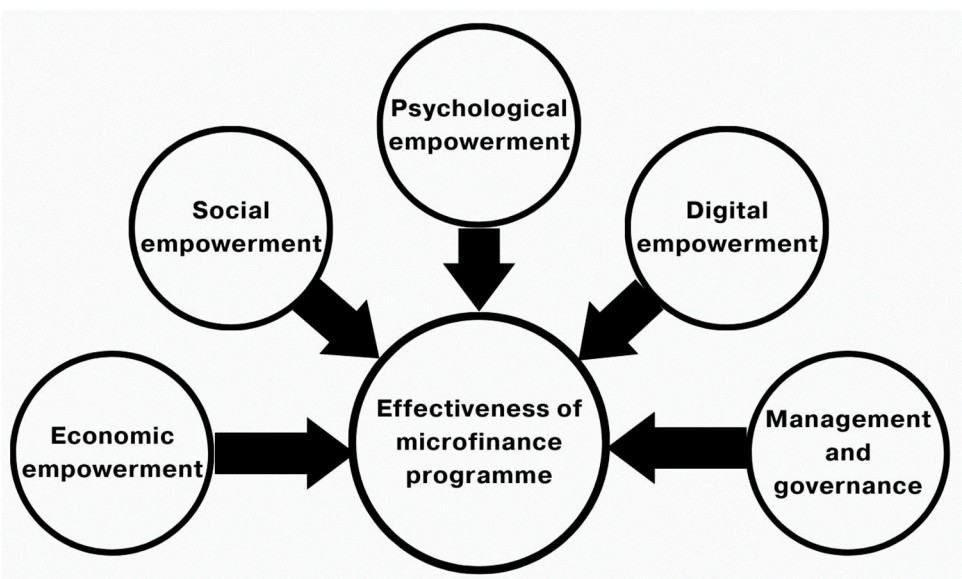

**Figure 1.** Antecedents of microfinance programme effectiveness during COVID-19 pandemic.

#### 3.2. Variable Measurement

An independent variable (IV) is a variable that can be manipulated and changed, and measures the effect of manipulation on other variables. Based on previous studies, to measure the empowerment of single mother entrepreneurs of microfinance participants, this study used four (4) leading indicators: economic, social, digital, and psychological. These four empowerment indicators are expected to impact and influence the effectiveness of the A.I.M. microfinance programme during the COVID-19 pandemic and MCO. For

this study, the dependent variable (DV) is the effectiveness of the A.I.M. microfinance programme during the pandemic and MCO. Table 1 presents the variables considered in this study, providing a comprehensive overview of the key factors under investigation The variables for this study are summarised in Table 1.

**Table 1.** Variables of the study.

| Variables Definition | Acronym | Sources |
|---|---|---|
| Dependent Variable: The effectiveness of the A.I.M. Microfinance Programme during the COVID-19 pandemic and the Movement Control Order (MCO) | E.M.F.P. | Ahmad et al. (2019) |
| Independent Variable: Economic Empowerment in an individual appraisal of household income, savings, cost of living, expenses, insurance coverage, business income, and business performance | EE | Ahmad et al. (2019). Al-shami et al. (2017a), Al-shami et al. (2017b) |
| Social Empowerment is an individual appraisal of social recognition, well-being, and interaction | SE | Ahmad et al. (2019), Ab-Rahim et al. (2018), Ahmad and Ahmad (2016) |
| Digital Empowerment is an individual appraisal of digital literacy, proficiency, and competencies | DE | Büchi (2020), Roffarello and Russis (2019), Diefenbach (2018) |
| Psychological Empowerment is an individual appraisal of self-confidence, self-skills, self-image, self-control, and mental well-being | P.E. | Ahmad et al. (2019), Salia et al. (2018), Ab-Rahim et al. (2018) |
| The management and governance factors of the A.I.M. microfinance programme cover aspects of administration, operation, and management | M.G.M.F.P. | Ahmad et al. (2019) |

### 3.3. Population and Sample Selection

The population refers to a group of individuals who want to be studied, namely single mother MSME entrepreneurs who participate in the A.I.M. microfinance scheme in the state of Selangor, the Federal Territory of Kuala Lumpur, and the Federal Territory of Labuan from 11 branches, namely, Ampang, Cheras, Kepong, Puchong, Sepang, Barat Selangor Sea, Hulu Selangor, Kuala Selangor, Shah Alam, Gombak, and Labuan. The single mother in question is a woman who (1) has had her husband die, (2) is divorced, or (3) is a woman whose husband is sick and unable to earn a living. Based on verified records and data obtained from the Research and Innovation Unit (U.P.I.), Amanah Ikhtiar Malaysia, the population of single mother entrepreneurs participating in the A.I.M. microfinance scheme who met the respondent criteria in these three states was 1677, so using the recommendations of Krejcie and Morgan (1970), the number of samples required was 314. This study used a quantitative method, where two sampling stages were used. The first stage was stratified sampling, where the total population was stratified according to 11 branches, and the percentage of each branch was obtained. The total sample for each stratum was also obtained using the same rate. Next, in the second sampling phase, respondents were randomly invited to respond to the distributed survey form.

### 3.4. Research Instrument

To obtain the necessary information, this research instrument used quantitative methods through questionnaires and interviews to collect data. Next, to increase the validity of

the indicators used in this study, the draft survey form was first submitted to the management of the Research and Innovation Unit (U.P.I.), Amanah Ikhtiar, Malaysia. The draft was also extended to managers, assistant managers and selected supervisors identified by U.P.I. Feedback, views, and remarks from both parties were considered in the design of the final survey form before being distributed to respondents. In addition, the research team also sought feedback from experts involved in entrepreneurship and management of MSMEs to further refine the indicators found in the survey form.

The design process of this structured questionnaire item is the result of a survey of previous studies. This questionnaire was used to collect primary data from a sample of single mother entrepreneurs of the A.I.M. microfinance programme, and is divided into the following nine sections: (A) Demographic aspects (respondent background); (B) the effectiveness of the A.I.M. microfinance programme and initiatives during the COVID-19 pandemic and MCO; (C) A.I.M. microfinance programme management and governance factors; (D) psychological empowerment; (E) economic empowerment; (F) social empowerment; (G) digital empowerment; (H) factors choosing the A.I.M. microfinance programme during the COVID-19 pandemic and MCO; and (I) open questions that gather general views as well as suggestions or feedback from respondents regarding the issue being studied.

Part A touches on the respondents' demographics, including household monthly income, marital status, family type, total household dependents, type of residence, residence status, and business location. Part B contains 11 indicators of the effectiveness of A.I.M. microfinance programmes and initiatives during the COVID-19 pandemic and MCO. Part C contains 13 indicators about the management and governance aspects of the A.I.M. microfinance programme. Part D includes 13 indicators of psychological empowerment. Next, Section E contains 12 indicators of economic empowerment. There are 11 indicators of social empowerment in Section F. This is followed by 11 indicators of digital empowerment in Section G. Next, in Section H, ten indicators are factors in selecting respondents to the A.I.M. microfinance programme during the COVID-19 pandemic and MCO. There is only one open question in Part I to allow respondents to give feedback freely about the issue or topic being studied. Each indicator was measured using four-point Likert scales for Sections B, C, D, E, F, and G. Researchers such as Ahmad et al. (2019) and Riduwan (2012) used four Likert scales. In fact, according to Garland (1991), the use of even response categories in the Likert scale eliminates the tendency of respondents to show a midpoint response that depicts a neutral reaction or implies non-decision to the surveyed item. The four-point Likert scales used in this study are represented by 1 = strongly disagree, 2 = disagree, 3 = agree, and 4 = strongly agree.

The average value (mean) was used to analyse the data. The mean score for each indicator was multiplied by a weighting of 2.5 to obtain an average value score (mean) based on a base score of 10. A weighting of 2.5 was used, considering this study used a four-point Likert scale, i.e., $2.5 \times 4 = 10$. Next, the average mean score for each domain was obtained by adding the mean score for all indicators with the total number of indicators in the domain in question. For example, there are 13 indicators for psychological empowerment because the mean score of the psychological empowerment domain will be obtained by summing up the mean scores of all empowerment indicators and dividing them by 13. Therefore, the analysis was performed based on the mean score table, which was categorised into four mean interpretations, which are less relevant, low, moderate, and high, based on a four-point Likert scale adapted from Ahmad et al. (2019) and Riduwan (2012), as shown in the following Table 2.

**Table 2.** Mean score interpretation.

| Mean Score | Interpretation |
|---|---|
| 2.50–3.75 | Less relevant |
| 3.76–6.25 | Low |
| 6.26–8.75 | Moderate |
| 8.76–10.00 | High |

*3.5. Data Collection Procedure*

As explained in the previous section, the data were collected using a questionnaire. Both primary and secondary data sources were used to complete the study. The primary data were obtained through a questionnaire distributed through the WhatsApp application and a Google Form link to single mother entrepreneurs participating as respondents in A.I.M. microfinance. In contrast, secondary data were obtained from various academic reference sources, journals, articles, documents, newspaper reports, and other relevant printed sources. Meanwhile, the primary information of single mothers as respondents, such as full name, identity card number, residential address, and telephone number, was requested and obtained from A.I.M.'s Research and Innovation Unit (U.P.I.). To verify the information of single mothers, whether active or inactive, the research team requested assistance from each branch manager and assistant branch manager, as well as supervisors in the three states.

For collecting information and research data, the list of respondents verified by U.P.I. and the branch office was submitted to the research team to form a WhatsApp group based on the branches. Eleven WhatsApp groups covering all branches involved in the study were created. Each research team member was placed in a WhatsApp group of two branches to distribute and obtain feedback on the questionnaire. The relevant research team members were also be responsible for sending reminders occasionally to improve the feedback and response rate from respondents who needed assistance in using Google Forms; research team members helped through telephone interviews based on the questions on the questionnaire.

*3.6. Data Analysis Techniques*

Data analysis techniques are a method of processing and presenting data and statistical procedures to make them easy to understand and provide solutions to research objectives. All of the data obtained were totalled and analysed with the help of S.P.S.S. 23. Data analysis was conducted in two (2) ways: descriptive analysis and inferential analysis. Descriptive analysis summarised the data numerically presented them using tables, graphs, or diagrams. These tables contain items, scores, frequency percentages, and means obtained from the responses of the study respondents. The inferential analysis aimed to test the hypotheses and interpret and produce conclusions based on correlation coefficient and multiple linear regression. The empirical model used in this study is as follows:

$$EMFPMi = \alpha + \beta1EEi + \beta2SEi + \beta3DEi + \beta4PEi + \beta5MGMFPi + \varepsilon i \qquad (1)$$

In this empirical model, the dependent variable is microfinance programme effectiveness or *E.M.F.P.* The independent variable consists of four indicators of empowerment of microfinance single mother MSME entrepreneurs, including economic empowerment (*E.E.*), social empowerment (*S.E.*), digital empowerment (*D.E.*), and psychological empowerment (*P.E.*). Then, *M.G.M.F.P.* is the management and governance factor of the microfinance programme, while $\varepsilon$ is the error term.

## 4. Results and Discussion

This section discusses the research results. Analysis and interpretation are presented to answer the underlying research questions. The results of these findings are divided into three parts: the analysis of the respondents' demographic information, descriptive analysis, and inferential analysis. This study received 422 responses.

*4.1. Respondents Profile*

Table 3 describes the profile of the study respondents based on selected sociodemographic elements, including business location, monthly household income, total dependents, respondent status, residential status, residential type, and family status. Regarding monthly household income, about 49.8 per cent of respondents earn less than MYR 2500. It

indicates that while the microfinance programme has successfully targeted and assisted the right group, more efforts are needed to raise respondents' economic well-being, particularly after the COVID-19 pandemic. The lower monthly income limits the ability of single mother MSME entrepreneurs to access better resources, including devices, digital technologies, and competencies enhancement opportunities post-COVID-19 pandemic, as suggested by Hamdan et al. (2021).

**Table 3.** Respondents' profile (indicator of selected sociodemographic attributes); N = 422.

| Demographic Attributes | Frequency | Percentage (%) |
|---|---|---|
| Business Location | | |
| Selangor | 301 | 71.3 |
| Wilayah Persekutuan—Kuala Lumpur | 99 | 23.5 |
| Wilayah Persekutuan—Labuan | 13 | 3.1 |
| Wilayah Persekutuan—Putrajaya | 9 | 2.1 |
| Monthly household income | | |
| B40 | | |
| Less than MYR 2500 | 210 | 49.8 |
| 3170–3169 | 105 | 24.9 |
| 3170–3969 | 40 | 9.5 |
| 3970–4849 | 24 | 5.7 |
| M40 | | |
| 4850–5879 | 20 | 4.7 |
| 5880–7099 | 6 | 1.4 |
| 7100–8699 | 4 | 0.9 |
| 8700–10,959 | 7 | 1.7 |
| T20 | | |
| 10,960–15,039 | 4 | 0.9 |
| More than 15,040 | 2 | 0.5 |
| Number of dependents | | |
| 1 to 4 | 303 | 71.8 |
| 5 to 9 | 4 | 0.9 |
| More than 10 | 77 | 18.2 |
| No dependents | 38 | 9 |
| Respondent status | | |
| Divorcee | 230 | 54.5 |
| Death of husband | 133 | 31.5 |
| Taking over husband's roles | 59 | 14 |
| Accommodation status | | |
| Poor People Housing Project | 9 | 2.1 |
| People Housing Project | 40 | 9.5 |
| Bungalow | 3 | 0.7 |
| Twin | 4 | 0.9 |
| High-rise | 121 | 28.7 |
| Village | 114 | 27 |
| Single-storey house | 58 | 13.7 |
| Double-storey house | 73 | 17.3 |
| Family status | | |
| Mixed | 6 | 1.4 |
| Dyad | 2 | 0.5 |
| Single parent | 215 | 50.9 |
| Cohabitation | 23 | 5.5 |
| Extended | 14 | 3.3 |
| Nucleus | 162 | 38.4 |

### 4.2. Descriptive Analysis

Table 4 shows a descriptive analysis of variables deployed in this study. On average, the respondents rated the effectiveness of the microfinance programme in assisting single

mother MSME entrepreneurs during the pandemic as moderate, with a mean score of 8.054. In terms of the microfinance programme's effectiveness contributing factors, the highest score is social empowerment with a mean score of 8.903, followed by digital empowerment with an 8.208 mean score, management and governance of microfinance programme with an 8.073 mean score, psychological empowerment with an 8.069 mean score, and finally, economic empowerment with a 7.649 mean score.

**Table 4.** Descriptive analysis of economic empowerment (N = 422).

| No | Variables | Min | Max | SD | Mean |
|:---:|:---:|:---:|:---:|:---:|:---:|
| 1 | Effectiveness of Microfinance Programme (E.M.F.P.) | 2.50 | 10.00 | 1.6347 | 8.054 |
| 2 | Economic Empowerment (E.E.) | 3.75 | 10.00 | 1.448 | 7.649 |
| 3 | Social Empowerment (S.E.) | 2.75 | 10.00 | 1521 | 8.903 |
| 4 | Digital Empowerment (D.E.) | 3.64 | 10.00 | 1.351 | 8.208 |
| 5 | Psychological Empowerment (P.E.) | 4.23 | 10.00 | 1.423 | 8.069 |
| 6 | Management and Governance of Microfinance Programme (M.G.M.F.P.) | 3.27 | 10.00 | 1.358 | 8.073 |

The Cronbach's alpha for all variables in the study exceeds a threshold of 0.70, as suggested by Nunnally and Bernstein (1994), thus indicating a reliable measurement. The Cronbach's alpha for the effectiveness of the microfinance programme is 0.954, for economic empowerment it is 0.910, for social empowerment it is 0.925, for digital empowerment it is 0.930, for psychological empowerment it is 0.945, and for management and governance of the microfinance programme it is 0.928

### 4.3. Empirical Assessment

Table 5 shows the results of the correlation analysis between dependent variables and independent variables, as well as the relationship between independent variables and other independent variables. According to Guilford (1973), the correlation or relationship between one variable and another variable can be classified as weak when the r value or correlation coefficient is between 0.20 and 0.40, at a moderate level between 0.40 and 0.70, at a high level between 0.70 and 0.90, and at a high level when the value of r exceeds 0.90. Meanwhile, if the r value is less than 0.20, the relationship between one variable and another can be ignored.

**Table 5.** Correlation analysis between variables (N = 422).

| | EMFP | M.G.M.F.P. | P.E. | E.E. | S.E. | D.E. |
|:---:|:---:|:---:|:---:|:---:|:---:|:---:|
| EMFP | 1 | | | | | |
| M.G.M.F.P. | 0.714 ** <0.001 | 1 | | | | |
| P.E. | 0.490 ** <0.001 | 0.550 ** <0.001 | 1 | | | |
| E.E. | 0.571 ** <0.001 | 0.623 ** <0.001 | 0.728 ** <0.001 | 1 | | |
| S.E. | 0.418 ** <0.001 | 0.563 ** <0.001 | 0.595 ** <0.001 | 0.843 ** <0.001 | 1 | |
| D.E. | 0.436 ** <0.001 | 0.470 ** <0.001 | 0.538 ** <0.001 | 0.612 ** <0.001 | 0.608 ** <0.001 | 1 |

** The correlation is significant at the 0.01 (2-tailed) stage.

Based on the results of the correlation analysis, it was found that the dependent variable—the effectiveness of the microfinance programme—has a high correlation (r = 0.714) with the management and governance factor and a moderate correlation with psychological empowerment (r = 0.490), economic empowerment (r = 0.571), social empowerment (r = 0.418), and digital empowerment (r = 0.436); all these correlations are significant at the $p = 0.01$ level. The analysis also shows that the correlation between the independent variable and other independent variables is also significant at the $p = 0.01$ level, however, with varying degrees of correlation. For example, the correlation between social and economic empowerment is high (r = 0.843). Meanwhile, the correlation between digital empowerment and management and governance factors is moderate (r = 0.470).

Table 6 reports the results of linear regression analysis on the relationship between the effectiveness of the microfinance programme and the indicators of empowerment of single mother entrepreneurs participating in the A.I.M. microfinance from an economic, social, digital, and psychological perspective as well as management and management aspects. Table 7 presents the results of the ANOVA. The multicollinearity analysis uses the variance inflation factor (V.I.F.) method (see Table 8). Multiple linear regression analysis shows that the two empowerment factors—economic and digital—are positive and significant at 1% and 5%, respectively. Thus, the results support the first and fourth hypotheses of the study. These finding are consistent with the previous study by Ahmad et al. (2019) on the positive and significant influence of economic empowerment towards the effectiveness of microfinance programmes in alleviating the well-being of women MSME entrepreneurs. Moreover, these results corroborate the earlier findings by Sri Rahayu et al. (2023), Mohammad et al. (2022), Hamdan et al. (2021), and Che Omar et al. (2020) on the need for digital empowerment for MSME entrepreneurs during the COVID-19 pandemic. Meanwhile, the social empowerment factor is negative and significant at 1%. The psychological empowerment coefficient was found to be positive but not significant. Hence, these results do not support the study's second and third hypotheses. These results contradict the earlier findings of Ahmad et al. (2019) on the positive and significant relationship of social and psychological empowerment towards the effectiveness of the microfinance programme. Meanwhile, the management and governance factor is positive and significant at 1%. Therefore, this result supports the fifth hypothesis of the study. The result is consistent with the finding of Ahmad et al. (2019) on the positive and significant influence of management and governance indicators towards the effectiveness of microfinance programmes. In conclusion, the results indicate that economic empowerment, digital empowerment, and management and governance aspects are positively and significantly related to the effectiveness of microfinance programmes in assisting women MSME entrepreneurs during the pandemic. In addition, the results do not find a positive and significant relationship between social empowerment and psychological empowerment towards the effectiveness of microfinance programmes in aiding women MSME entrepreneurs during the COVID-19 outbreak. We believe that the absence of a positive and significant relationship between social empowerment and psychological empowerment towards the effectiveness of microfinance programmes is influenced by respondents' situational and contextual experiences during the COVID-19 pandemic. During the pandemic, social interaction was limited due to movement restrictions ordered by the government to curb the spread of the lethal virus. Moreover, the psychological aspect of respondents during the crisis period differs from the non-crisis time. Furthermore, from Table 7, the ANOVA results indicate the significance of the model of the study. In addition, the degree of multicollinearity in the regression model is quantified by the variance inflation factor (V.I.F.). In a multivariate regression model, multicollinearity occurs when there is a high correlation between several of the independent variables. Table 8 presents the Variance Inflation Factor (V.I.F.) results, indicating that all V.I.F.s are below 10, suggesting the absence of multicollinearity issues in the model analysis conducted for this study. The regression model utilized in the study is expressed as follows:

**Table 6.** Multiple linear regression results (N = 422).

| Item | Results |
|---|---|
| $\alpha$ | 0.825 |
| MGMFP | 0.721 *** |
| PE | 0.0000 |
| EE | 0.460 *** |
| S.E. | −0.035 *** |
| DE | 0.127 ** |
| Adjusted R$^2$ | 0.566 |

** Significant at 0.05; *** significant at 0.01.

**Table 7.** Analysis of variance (ANOVA) [a] results.

| Model | Sum of Squares | df | Mean Square | F | Sig. |
|---|---|---|---|---|---|
| Regression | 630.870 | 5 | 126.174 | 107.268 | <0.001 [b] |
| Residual | 483.441 | 411 | 1.176 | | |
| Total | 1114.311 | 416 | | | |

[a] Dependent variable: E.M.F.P; [b] Predictors: (Constant), E.E., S.E., PE, DE, M.G.M.F.P.

**Table 8.** Variance inflation factor (V.I.F.) results.

| Item | VIF |
|---|---|
| MGMFP | 1.725 |
| PE | 2.249 |
| EE | 4.934 |
| SE | 3.632 |
| DE | 1.759 |

## 5. Conclusions

This subsection deliberates the achievement of research objectives, managerial and practical implications, limitations, suggestions for future studies, and summarises the current study.

### 5.1. Achievement of Research Question
Research Question

What is the relationship between the four empowerment indicators of single mother MSME entrepreneurs—economic, social, digital, and psychological—as well as the governance aspects and the effectiveness of microfinance programmes when facing the COVID-19 pandemic and MCO?

The results of the multiple linear regression analysis find evidence and support towards three of the formulated hypotheses of this study. Economic empowerment, digital empowerment, and the management and governance of microfinance programmes have a positive and significant relationship with the effectiveness of microfinance programmes in assisting single mother MSME entrepreneurs during the COVID-19 pandemic and MCO.

### 5.2. Theoretical Implications

The study's outcomes add to the body of knowledge, particularly enhancing the application of empowerment theory in explaining the role of digital empowerment as a tool to achieve the effectiveness of microfinance programmes in aiding women MSME entrepreneurs in surviving the crisis. The findings also explain empowerment theory perspectives on the empowering process of individuals, i.e., women MSME entrepreneurs, in terms of economic and digital processes as well as the organisational empowering process, i.e., management and governance aspects achieved the desired outcome, which is the effectiveness of microfinance programmes in assisting women MSME entrepreneurs during the challenging period. In addition, the findings further consolidate previous studies'

perspectives on the need to empower economic and digital aspects towards achieving the effectiveness of microfinance programmes in the crisis period. In addition, a continuous effort to strengthen the management and governance of the microfinance programme is essential to achieve the desired effectiveness in assisting targeted participants in overcoming crisis hardships.

### 5.3. Managerial and Practical Implications

Several policy recommendations are highlighted to increase the capacity and effectiveness of the microfinance scheme, particularly for future strategies in response to COVID-19 challenges. These recommendations aim to prepare the target group better for future shocks or crises.

### 5.3.1. Management and Governance of the Microfinance Programme

A comprehensive, complete, updated, and networked database with relevant ministries, agencies, or departments should be worked on so that financial and non-financial assistance can be delivered more effectively and systematically, and in a targeted manner. The characteristics of such a database will enable the implementation of more effective and systematic monitoring, making it easier for interested parties to access information accurately and subsequently channel the necessary support more quickly, thoroughly, and intelligently.

### 5.3.2. Economic Empowerment

Continuous awareness campaigns are needed to create awareness among microfinance participants that they should not rely too much on borrowing practices. A loan that cannot be managed well will eventually harm the borrower. This matter can be strengthened with training, business skills and guidance, market research, financial administration management, and financial literacy that can have a significant impact on improving living standards and the process of empowerment, especially for single mother entrepreneurs, and the economic development of the state and the country. In addition, the implementation of targeted aid initiatives for microfinance participants should be strengthened. This targeted assistance method allows microfinance agencies to channel financial and non-financial assistance based on the participant's specific needs or challenges. Based on the principle of 'the right medicine for the specific disease', this approach allows the participants to overcome the challenges they face more thoroughly and quickly. This approach increases economic development and cost-effectiveness.

### 5.3.3. Digital Empowerment

Continuous and structured training and digital skills improvement programmes should be carried out, especially with Malaysia stepping into the digital economy era. Participants should be widely exposed to digital marketing techniques, digital payments, and other online transactions. In addition, continuous efforts are needed to increase accessibility to the Internet and gadgets among single mother entrepreneurs.

### 5.3.4. Social Empowerment

Multi-directional cooperation should be strengthened across agencies and multi-stakeholders to increase the social empowerment efforts of single mother entrepreneurs. A transparent and comprehensive multilateral and multi-agency cooperation framework is needed to avoid duplication of functions, waste of resources, and leakage of funds. In addition, multi-stakeholder collaboration is essential to establish a 'mentoring' system and 'networking' programme for microfinance participants, especially for single mother entrepreneurs. To ensure that the concerned group continues to benefit for a long time, the mentoring system and networking programme are essential because they can significantly impact living standards, the empowerment process, and the development of the country's economy.

### 5.3.5. Psychological Empowerment

Mental and psychological well-being should be emphasised to ensure that single mother entrepreneurs participating in microfinance programmes have good mental health. Continuous guidance from certified counsellors and easily accessible health experts should be considered, since single mother entrepreneurs often face various economic, social, and family management challenges in their self-development.

### 5.4. Limitations of Study

Although this study achieved the research objectives outlined, there are at least two limitations or constraints. First, this study concentrated on respondents in urban areas of three states, covering only eleven branches. Second, the response of this study focused on a group of vulnerable communities consisting of single mother MSME entrepreneurs among microfinance participants.

### 5.5. Suggestions for Future Research

Based on the research that has been conducted and considering the respondents' views, we would like to put forward some suggestions that can be considered for implementation in the future. Among them, (1) the study in the future can be expanded to involve respondents in other states, because it consists of the analysis of more branches. Next, a comparative analysis regarding the effectiveness of microfinance programmes or initiatives during the crisis can be refined between states, branches, and localities such as urban–rural–suburban areas; (2) future studies can also be expanded by involving more respondents from other vulnerable communities among microfinance participants, such as disabled entrepreneurs, communities in public housing projects, and youth from poor urban and rural areas; and (3) future studies can also consider a longitudinal study approach to enable the assessment of the empowerment factors of microfinance participants and the effectiveness of the microfinance programme for a more extended period compared to cross-sectional studies.

### 5.6. Summary

Overall, this study identified the effectiveness of microfinance programmes and initiatives during the COVID-19 pandemic and MCO from the perspective of single mother MSME entrepreneurs who participate in the programme. In addition, the study identified the relationship between the four empowerment indicators of single mother entrepreneurs participating in the microfinance programme—economic, social, digital, and psychological—and the programme's management and governance factors. Differentiating from Ahmad et al. (2019), this current study found evidence of the importance of digital empowerment as one of the forward-moving elements for single mother entrepreneurs' success in the COVID-19 period. In addition, the findings from this study established empirical support for Mohammad et al. (2022), Hamdan et al. (2021), and Che Omar et al. (2020), and their qualitative and conceptual analyses on appropriate mitigation and forward-looking strategies for MSME entrepreneurs' post-pandemic era, including among women entrepreneurs of microfinance programme participants. The study's findings provide a clear picture to all interested parties, especially the microfinance agency and the government, in supporting the well-being of single mother MSME entrepreneurs participating in the microfinance programme. Generally, change, reform, and progress in a society begin with women, because this group is the cornerstone of a family institution. To succeed, the microfinance programme must be supported by several other aspects, such as training, skills, and educational opportunities for single mothers, especially to create opportunities to obtain digital lifelong learning. Various development activities, technical and vocational training, and targeted comprehensive financial and non-financial aid initiatives must be provided to increase women's empowerment, especially among single mother MSME entrepreneurs. Resilient women can transform families, changing society and the economy.

**Author Contributions:** Conceptualisation, A.S.A.S. and N.M.Y.; methodology, S.M.Z.; validation, A.S.A.S., N.M.Y. and S.M.Z.; formal analysis, A.S.A.S. and S.M.Z.; resources, A.S.A.S., N.M.Y. and S.M.Z.; writing—original draft preparation, A.S.A.S.; writing—review and editing, N.M.Y.; supervision, A.S.A.S.; project administration, N.M.Y.; funding acquisition, N.M.Y. All authors have read and agreed to the published version of the manuscript.

**Funding:** This research was funded by the National Population and Family Development Board Malaysia (N.P.F.D.B.), research grant GPLPPKN0294, and the Multimedia University Internal Research Fund 2022, M.M.U.I./220005. Multimedia University, Malaysia, funded the A.P.C.

**Informed Consent Statement:** Not applicable.

**Data Availability Statement:** Data are not publicly available due to copyright.

**Acknowledgments:** The team acknowledges and expresses gratitude for the funding assistance from the National Population and Family Development Board Malaysia (N.P.F.D.B.) (GPLPPKN0294) and the Multimedia University's Internal Research Fund (M.M.U.I./220005). The team also would like to thank Baktiar Hasnan, Nur Rabikha Zainudin, and Hani Suhaila Ramli from PENDIDIK (Association for Digital Community Education) for assisting with the data collection process and the logistical support rendered by the Research and Innovation Unit of Amanah Ikhtiar Malaysia (A.I.M.) towards the completion of research.

**Conflicts of Interest:** The authors declare no conflicts of interest.

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
