# Peer review of "Examining the Effects of the Pandemic on Entrepreneurial Activities among Urban Single Mothers: An Exploratory Study"

_ijfs, doi:10.3390/ijfs12020030_

Round 1
Reviewer 1 Report
Comments and Suggestions for Authors
Please, avoid using acronyms.
Author Response
Dear Reviewer,
Thank you for the update and for providing the feedback regarding our manuscript with the ID: ijfs-2848740. We appreciate the time and effort the reviewers have dedicated to evaluating our work.
We have carefully reviewed the referee reports and noted the suggested revisions. We ensure that all comments are addressed comprehensively in the revised manuscript. Additionally, we have attached a cover letter detailing our responses to each point raised by the referees, as requested.
Once again, thank you for your guidance throughout this process.

Reviewer 2 Report
Comments and Suggestions for Authors
The paper "Surviving the Storm: An Exploratory Analysis of Pandemic's Impact on Urban Single Mother Entrepreneurship" examines COVID-19's effects on single mother entrepreneurs in the Amanah Ikhtiar Malaysia (AIM) microfinancing program. the manuscript requires significant revisions to better align with scholarly standards and enhance its academic contribution.
1. The abstract and other sections contain repetitive information more suited to an introduction, which dilutes the paper's focus and impact. The abstract should concisely summarize the study's objectives, methodology, key findings, and scholarly contributions without introductory filler.
2. There is an excessive emphasis on descriptive and demographic details that do not advance theoretical or empirical discussions, limiting the paper's contribution to the literature. This needs to be eliminated and prioritize the regression analysis, providing a thorough examination of its implications for microfinancing effectiveness during crises. This should be the focal point for demonstrating the study's contribution to the field.
3. Deepen the engagement with existing literature to contextualize findings within broader theoretical frameworks, particularly focusing on entrepreneurship, for example> An empirical study of entrepreneurial leadership and fear of COVID-19 impact on psychological wellbeing: A mediating effect of job insecurity, T Guberina, AM Wang and
Walid, L. & Peng, H. (2022). Entrepreneurial Risk Perception and Sustainable Entrepreneurship Intention among SMEs in Algeria: A Multidimensional Approach. Journal of Entrepreneurship and Business Development, 2(2), 7-15.
Ilieva-Koleva, D. & Dobreva, J. (2021). Social Entrepreneurship as a Form of Social Responsibility in Bulgaria. International Journal of Operations Management, 1(3), 25-31.
4. The regression model and hypotheses, crucial to the paper's aim, are overshadowed by descriptive content. Focusing on and elaborating these aspects would substantially strengthen the study.
The discussion section is non existent, there is no intepretation in context of past studies. Overall the paper needs significant editing and improvement
Comments on the Quality of English LanguageThe paper needs improving.
Author Response
Dear Reviewer,
Thank you for the update and for providing the feedback regarding our manuscript with the ID: ijfs-2848740. We appreciate the time and effort the reviewers have dedicated to evaluating our work.
We have carefully reviewed the referee reports and noted the suggested revisions. We ensure that all comments are addressed comprehensively in the revised manuscript. Additionally, we have attached a cover letter detailing our responses to each point raised by the referees, as requested.
Regarding the English revisions, we have used a specialised tool, as suggested, to ensure that the language meets the required standards. We are confident that the revised version will meet the criteria needed for publication.
Once again, thank you for your guidance throughout this process.

Reviewer 3 Report
Comments and Suggestions for Authors
The manuscript had a limited importance for scientific community because it refers to a single country.
The title of the article is not suitable. The title of an article should be simple, concise, without sensational expressions. Therefore, the metaphor "surviving the storm" is beautiful but entirely unsuitable for a scientific article title. We propose the straightforward title "An Exploratory Analysis of Pandemic's Impact on Urban Single Mother Entrepreneurship."
The scientific literature review needs revision; there are too few bibliographic references to the pandemic. Additionally, we do not understand how references to the pandemic with articles from 2016 can be made in the introduction. For example, "For MSMEs, the impact of this pandemic has had a big impact on their business activities (line 42), and Furthermore, Bartz and Winkler (2016) emphasize that MSME (line 45)." The introduction needs to be reworked.
The statistical analysis is detailed, especially in terms of descriptive statistics, but there is insufficient coverage of inferential statistical analysis. Although an empirical model is appropriately selected (line 457), the analysis of this model is not delineated (not just descriptive analysis in tables 4-11). We recommend revising the statistical analysis section.
Comments on the Quality of English LanguageThe quality of English language must be improved. Since the authors do not appear to be native English speakers, I suggest a comprehensive revision of the English language and style for the entire article using Grammarly or another specialized tool
Author Response
Dear reviewer,Thank you for the update and for providing the reviewer feedback regarding our manuscript with the ID: ijfs-2848740. We appreciate the time and effort the reviewers have dedicated to evaluating our work. We have carefully reviewed the referee reports and noted the suggested revisions. We will ensure that all comments are addressed comprehensively in the revised manuscript. Additionally, we have attached a cover letter detailing our responses to each point raised by the referees, as requested. Regarding the English revisions, we have used a specialised tool as suggested to ensure that the language meets the required standards. We are confident that the revised version will meet the criteria needed for publication. Once again, thank you for your guidance throughout this process.

Round 2
Reviewer 2 Report
Comments and Suggestions for Authors
edit and improve the paper to be more coherent, and also need to rephrase hypotheses , what kind of relationship is suggested, positive negative, etc. it is not correcrly phrased, and entire theory should suite the hypotheses
Comments on the Quality of English Languageproofread further
Author Response
Dear reviewer,
We sincerely appreciate your time and effort in reviewing our manuscript “Examining the Effects of the Pandemic on Entrepreneurial Activities among Urban Single Mothers: An Exploratory Study”. Your insightful comments and positive feedback have been precious in refining our work.
We are pleased to inform you that we have carefully considered your suggestions and incorporated several improvements into the paper. These enhancements include:
- The direction of the hypothesis. Amended and improved affected results, findings, discussions and conclusions.
- Improve the cohesiveness of the paper (Abstract to Reference)
Once again, thank you for your positive review and for helping us improve our manuscript.
Reviewer 3 Report
Comments and Suggestions for Authors
All the modifications suggested by the reviewers have been implemented by the authors, significantly improving the quality of the article. We believe that in its current form, the article meets the requirements for publication in the journal.
Author Response
Dear Reviewer,
Thanks to your constructive feedback, we are confident that our paper is now in a much stronger position for publication. We are grateful for your support and await your continued guidance.
Thank you.